# Second Harmonic Generation Versus Linear Magneto-Optical Response Studies of Laser-Induced Switching of Pinning Effects in Antiferromagnetic/Ferromagnetic Films

Irina A. Kolmychek [1], Vladimir B. Novikov [1], Nikita S. Gusev [2], Igor Yu. Pashen'kin [2], Evgeny A. Karashtin [2] and Tatiana V. Murzina [1,*]

1    Physics Department, M. V. Lomonosov Moscow State University, 119991 Moscow, Russia; irisha@shg.ru (I.A.K.)
2    Institute for Physics of Microstructures, RAS, 603087 Nizhny Novgorod, Russia
*    Correspondence: mur@shg.ru

**Abstract:** Composite magnetic nanostructures are a subject of high research interest, as they provide a number of exciting effects absent in live nature. Among others, much attention has been paid to the studies of exchange coupling in antiferromagnetic/ferromagnetic (AFM/FM) films, which leads to the pinning effect. It manifests itself as a widening and shift of the magnetic hysteresis loop with respect to zero value of the external magnetic field oriented along the pinning direction. In this work, we report on comparative studies of linear and nonlinear magneto-optical effects under the laser-induced switching of the pinning effect in IrMn/CoFe films of various thickness of the ferromagnetic CoFe layer. We found that the magneto-optical response of the pinned AFM/FM nanofilms appears with different hysteresis loop parameters in the transverse magneto-optical Kerr effect (MOKE) and interface-sensitive magnetization-induced second harmonic generation (SHG), indicating the diversity of the magnetic effects at interfaces compared to the bulk of the films.

**Keywords:** exchange bias; pinning; magneto-optics; second harmonic generation

## 1. Introduction

Modern technologies enable the production of high-quality metal magnetic films with nanometer thickness offering new possibilities for the development of novel magnetic and transport phenomena in artificial nanomaterials [1]. Fascinating effects appear when different types of magnetic materials are composed in a single structure. Indeed, great research endeavors are aimed towards harnessing the magnetic interaction between adjacent antiferromagnetic (AFM) and ferromagnetic (FM) layers, where a proper composition results in the exchange coupling of spins of the two constituent materials. This leads to the appearance of the unidirectional magnetic anisotropy, i.e., to the pinning effect [2,3].

Interface exchange coupling can arise after cooling a thin AFM/FM bilayer in the presence of a static magnetic field **H** from the temperature exceeding the Neel temperature $T_N$ of an antiferromagnetic film, which in turn should be lower than the Curie temperature $T_C$ of the FM layer. After this, at the temperature $T < T_N$ the AFM/FM system exhibits the magnetic hysteresis loop shifted in the direction opposite to the magnetic field applied during the cooling process [2–5]. The following intuitive physical picture can be exploited to explain this effect. For the AFM/FM layered system subjected to the procedure described above, the magnetization reversal of the FM layer requires higher **H** than in the absence of the pinning effect as it has to overcome the microscopic torque exerted by ordered spins at the interface with the AFM film. Thus, the unidirectional magnetic anisotropy is formed, the main attribute of which is the magnetic hysteresis loop asymmetry with respect to the zero value of the external magnetic field applied parallel to the pinning direction. The accompanying hysteresis loop shift is known as the exchange bias field, $H_{EB}$.

It has also been recognized that the treatment of an AFM layer using the described thermal-assisted approach results in the magnetic anisotropy even in the absence of an adjacent ferromagnetic film, while its value can be lower as compared to that for the AFM/FM bilayer made under similar conditions [6]. In the case of AFM/FM layers with low AFM anisotropy, the field cooling results in broadening the magnetic hysteresis loop, i.e., it leads to an increase in coercivity $H_C$ as compared to that of the same structure prior to the field cooling. The mechanism of this effect is the rotation of spins of both the FM and AFM layers under the reversion of the external magnetic field that requires more energy and thus broadens the magnetic hysteresis loop [3].

Importantly, the magnetic interface coupling is rather sensitive to the temperature of an AFM/FM structure [7,8]. Typically, the exchange bias in the AFM/FM films breaks down at temperatures exceeding $T_N$, while in nanostructured systems this coupling has been shown to disappear at temperatures much lower than $T_N$ due to size effects. The temperature at which the hysteresis loop shift vanishes is denoted as blocking temperature $T_B$ [9,10].

A promising method for optical control of the pinning effect is laser-induced heating of the AFM/FM interface; namely, as the structure is heated above the blocking temperature, the pinning is destroyed. This provides a possibility to switch this effect and thus to control the magnetic interaction between the FM and AFM layers [3,11]. A complete reversal of the exchange bias at the AFM/FM interface can be induced as well by a single femtosecond pulse of sufficiently high energy [12–17]. Spin precession in magnetic structures subjected by ultrashort excitation provides extensive information on magnetic interactions that was harnessed for the studies of the exchange bias in pinned films by ultrafast pump-probe [18] and terahertz emission spectroscopy [11]. Laser pulse perturbation is found to modify the exchange bias as evidenced by the shift of the hysteresis loop with respect to the applied DC magnetic field as observed by the magneto-optical Kerr effect [11]. The modification of the exchange bias by means of intense laser beam irradiation of the magnetic AFM/FM structure looks attractive as the spatial region of this impact is comparable to the laser beam spot, which can be used for optical data writing and storage [13,19–23].

The optical second harmonic generation (SHG) technique is known as a sensitive tool for studies of interfaces and nanostructures made of materials with inversion symmetry. Moreover, when being applied to studies of magnetic structures, magneto-optical effects at the SHG wavelength appear with the typical values being one–two orders of magnitude higher compared to their linear analogues such as magneto-optical Kerr effect (MOKE) [24,25]. This nonlinear optical method allows one to distinguish various types of nontrivial magnetic states, such as magnetic vortices and magnetic toroidal moment, gradients of magnetization, etc. [26–29]. At the same time, to the best of our knowledge, no detailed studies of the SHG effect in the exchange-biased AFM/FM systems have been performed up to now. In this paper, we present the results of the experimental studies of linear and nonlinear optical response of a set of AFM/FM (IrMn/CoFe) structures with thickness of the FM layer from 2 nm up to 50 nm under the laser-induced switching of the pinning effect. The difference in linear and nonlinear optical responses of the structures is unveiled.

## 2. Materials and Methods

The experimental structures under study are planar bilayered IrMn/CoFe films with a thickness of the antiferromagnetic IrMn layer of 10 nm, and the thickness of the ferromagnetic CoFe layer ($d_{CoFe}$) ranging from 2 nm to 50 nm. The films were covered by a 2 nm Si protective film. These structures were deposited on a glass substrate by magnetron sputtering in an argon atmosphere at a pressure of $2 \times 10^{-3}$ Torr with a static in-plane magnetic field of about 130 Oe applied during the deposition of metals. All the structures were annealed at 220 °C for 90 min at low pressure ($10^{-5}$ Torr) when applying the DC magnetic field of 1500 Oe in the same direction as during the IrMn sputtering. As the blocking temperature of thin IrMn films is about $T_B = 200$ °C [9], the Curie temperature

of CoFe is $T_C$ = 977 °C [30]; the CoFe layer becomes pinned to the adjacent IrMn layer at room temperature after this treatment.

Experimental studies were performed using the setup shown schematically in Figure 1a. It was specially assembled to provide the magnetic hysteresis loop measurements by means of both MOKE and SHG methods. The former was measured using the halogen lamp as a light source, while the SHG studies were performed when using the radiation of a Ti-sapphire laser. Both p-polarized light beams were focused by a lens to the same area on the sample, allowing us to directly compare the linear and nonlinear response of the films. The angle of incidence was equal to 45 °. Laser-induced switching of the pinning of the AFM/FM film was performed by extra optical beam using Ti-sapphire (Ti-Sa) laser.

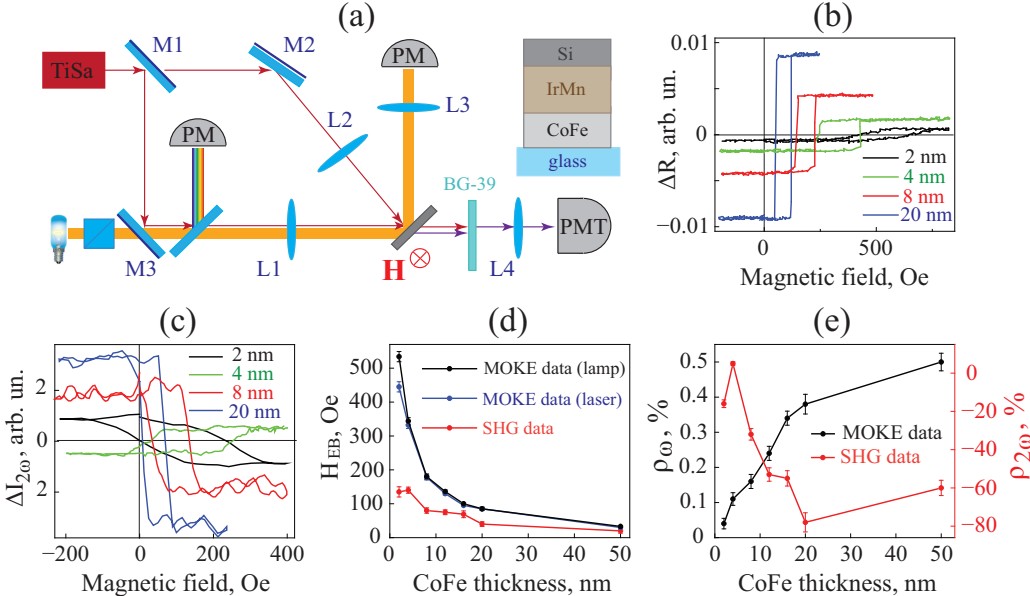

**Figure 1.** (**a**) Experimental setup and scheme of the structures under study; (**b**) transverse MOKE hysteresis loops; (**c**) SHG hysteresis loops; (**d**) dependence of the values of the hysteresis loops' shifts obtained from the MOKE and SHG data on CoFe thickness; (**e**) dependence of the magnetic contrast for the linear reflection coefficient (black points, left axis) and transmitted SHG intensity (red points, right axis).

In more detail, the setup operated as follows. For the linear MOKE measurements, the mirror M3 was flipped down, whereas the lens L1 focused the broadband light on the studied film into a spot with a diameter of 70–100 μm. Two photodiode power-meters (PMs) Thorlabs S121C (Newton, NJ, USA) were used as signal and reference detectors, so that the normalized power of light reflected from the sample was measured. The DC magnetic field of up to 1 kOe formed by a programmatically controlled electromagnet was applied to the films in the transversal geometry.

In the nonlinear optical experiments, a Ti-Sa laser was used as a source of fundamental radiation with a wavelength of 800 nm, the repetition rate of 80 MHz, and pulse duration of 30–50 fs. In this case, mirrors M1 and M3 were flipped up and the lens L1 focused the laser beam on the sample into a spot of about 20 μm in diameter. The second harmonic power generated in transmission through the sample was detected by a photomultiplier tube (PMT, Hamamatsu R4220, Shizuoka, Japan), while the transmitted light at the fundamental frequency was blocked by a color filter BG-39. The magnetization-induced SHG hysteresis loops were obtained for the average power of the incident laser beam of about 12–13 mW, which corresponds to the peak intensity of the laser pulse of 1 GW/cm². Applied light power ensured that the structure did not get hotter than the $T_B$ under the laser beam irradiation.

For the laser-induced switching of the pinning effect in the AFM/FM structure, the mirror M1 was flipped down and the heating Ti-Sa beam was focused at near-normal incidence by the lens L2 with the focal distance of 70 mm. This lens was placed on a two-coordinate motorized translator, the application of which allowed us to perform the laser heating for a square area of 100 μm × 100 μm. The beam had an average power of 55–60 mW and was focused into a spot of 30 μm in diameter, which corresponds to the mean light intensity of about 8.5 kW/cm$^2$ (peak intensity of about 3.5 GW/cm$^2$). It is crucially important that (i) the laser-treated region of the film is larger than the focal spot of the lamp and of the probe Ti-Sa light sources and (ii) all three beams (one from the halogen lamp and the two laser beams used for heating the sample and for the SHG measurements) were aligned in the same spot of the structure using a CMOS camera.

## 3. Results

### 3.1. Moke vs. SHG Magnetic Measurements

Firstly, we address the results of measurements of the transverse MOKE and SHG hysteresis loops for as-prepared IrMn/CoFe structures. For this, we acquired the magnetic field dependencies of the magnetization-induced change of (a) the reflection coefficient ($\Delta R$) and (b) the SHG intensity ($\Delta I_{2\omega}$) relative to the averaged SHG intensity value. It is necessary to outline that the nonlinear optical experiments were performed at low input laser intensity (12–13 mW) in order to prevent the influence of the laser radiation on the magnetic state of the film (see Section 3.2). The obtained hysteresis loops are shown in Figure 1b,c for MOKE and SHG, respectively, for various thicknesses of the CoFe layer, when light is incident on the sample from the IrMn side. We found that all dependencies $\Delta R(H)$ and $\Delta I_{2\omega}(H)$ are shifted towards the positive direction of the magnetic field $H$ whereas the values of the exchange bias and the width of the hysteresis loops vary with the ferromagnet thickness (Figure 1d). Namely, a monotonous decrease in $H_{EB}$ is observed with the increase in $d_{CoFe}$.

To check that the low-intensity laser beam used in the SHG measurements did not change the magnetic state of the studied structure, we performed additional MOKE measurements when irradiating the samples by the fundamental laser beam with the power of 12–13 mW. We used the mirrors M1 and M3 to direct the femtosecond laser beam towards the film and the two identical power-meters (PM) as detectors (Figure 1a). The values of $H_{EB}^{MOKE}$ obtained for this low-intensity laser irradiation are shown in Figure 1d, blue points. One can see that for all CoFe thicknesses the value of $H_{EB}^{MOKE}$ coincides with that obtained when using the halogen lamp as the probe source, except a small difference observed for the thinnest film with $d_{CoFe}$ = 2 nm, confirming the preservation of the magnetic properties of the samples during the SHG measurements.

We found that for all the structures in the series, the shift of the MOKE hysteresis loops is larger compared to that obtained in the SHG measurements, $H_{EB}^{SHG} < H_{EB}^{MOKE}$ (Figure 1d). This is probably attributed to different localization of the linear and nonlinear optical source terms within the films as is discussed below.

As a measure of the transversal magnetization-induced effects in the linear and nonlinear optical response, we used the magnetic contrast defined as:

$$\rho_{\omega,2\omega} = \frac{I_{\omega,2\omega}(+H) - I_{\omega,2\omega}(-H)}{I_{\omega,2\omega}(+H) + I_{\omega,2\omega}(-H)}, \tag{1}$$

where $I_{\omega,2\omega}(\pm H)$ are the reflected or transmitted light intensity at the fundamental or the SHG frequency obtained for the opposite directions of the saturated external magnetic field, $\pm H$. The obtained $\rho_\omega$ and $\rho_{2\omega}$ values (Figure 1e) are tenths and dozens of percent, respectively, which is typical for ferromagnetic metals [26]. The magnitudes of the magnetization-induced effects in linear and nonlinear optical response increase with the CoFe thickness (Figure 1e), indicating the growth of the volume of the magnetic material and of the relevant specific magnetization. Interestingly, $\rho_{2\omega}$ changes its sign with the

thickness of the FM layer; namely, it is positive for $d_{CoFe} = 4$ nm and negative for other structures from the series.

To investigate whether different interfaces play a role in the linear and nonlinear magneto-optical effect, similar experiments were performed when irradiating the structures from the substrate side. In this case, the optical beam first reached the glass side and the CoFe layer, and then passed through the IrMn/CoFe interface. It was found that the parameters of the hysteresis loops in both MOKE and SHG response coincide with those presented in Figure 1d within the experimental accuracy. When irradiating the structure from the substrate side, the values and sign of the magnetic contrast $\rho_\omega$ coincide with those shown by black points in Figure 1e, whereas the values of $\rho_{2\omega}$ are of the opposite sign to the data presented in Figure 1e by red points.

### 3.2. Laser-Induced Pinning Switching

The next stage of the study was devoted to the laser-induced switching of the pinning effect in the AFM/FM structures. Below, we present the experimental data for the IrMn/CoFe film with the CoFe thickness of 4 nm, while it was checked that the samples with all the other CoFe thicknesses from the considered series demonstrated qualitatively similar effects. The laser-induced switching experiment involved illuminating a 100 μm × 100 μm area of the AFM/FM film with an intense laser beam (see Section 2), resulting in the IrMn/CoFe bilayer being heated above the blocking temperature $T_B$. Subsequently, the illumination was turned off and the sample was cooled down to room temperature while exposed to the saturating magnetic field of 800 Oe. To illustrate different steps of this process, we sketched them in Figure 2a,b, where the alignment of spins of the AFM material for the two opposite directions of **H** are shown schematically. In such a scheme, the direction of the external magnetic field applied during the films' cooling defines the pinning direction.

Firstly, we found the critical power of Ti-Sa heating beam that leads to the pinning switching. The sample was positioned in such a way that the external magnetic field was applied along the pinning direction and perpendicular to the plane of the light incidence. The MOKE loop was obtained in the same film region before (Figure 2c, wine-colored points) and after the laser beam heating process performed for different average laser power (15–60 mW), with positive-in-sign magnetic field being applied during the consequent cooling process. It can be seen that irradiating the sample by the laser beam of the power 15 mW or less (red points) does not change the magnetic properties of the film. The power range of 15–45 mW (orange and green points) can be considered as transitional in the sense that the MOKE hysteresis loop of the films is modified as compared to that of the as-prepared sample. This gentle change instead of an abrupt one can be explained by the inhomogeneity of the magnetic properties of the sample caused by the laser exposure. Finally, using the laser beam power exceeding 45 mW and until thermal destroying of the structure, the reversal of the pinning effect was realized. It appears as the MOKE hysteresis loop undergoing a transfer that is nearly mirror-symmetric with respect to $H = 0$ due to the described laser impact (blue points, Figure 2c). It is worth noting that the same change behavior was observed for both the femtosecond and continuous wave regimes of the Ti-sapphire laser used for the pinning switching which indicates the thermal mechanism of this effect. Further, the switching of the pinning effect was realized using a light power of 60 mW.

Then, we demonstrate the multiple switching of the pinning effect for IrMn/CoFe films harnessing the MOKE and SHG response. Figure 2d–f shows consequently the MOKE hysteresis loops of the non-perturbed IrM/CoFe(4nm) film and after the laser-induced switching performed by 60 mW laser beam in the presence of the positive (Figure 2a) and negative (Figure 2b) magnetic field. Analogous SHG measurements performed under the same conditions were carried out and the corresponding results are shown in Figure 2g–i. Obtained SHG hysteresis loops were acquired using the fundamental beam with the power of 12–13 mW, which excluded its influence on the magnetic state of the samples still keeping measurable values of the transmitted second harmonic intensity. We found that high-intensity laser irradiation of the film region followed by cooling in the saturated

positive magnetic field induces the pinning switching (Figure 2a). It is sustained by the shift of the MOKE and SHG hysteresis loops towards the negative magnetic field (Figure 2e,h). The assumed spin alignments in the AFM/FM system in saturated magnetic fields are schematically illustrated at the top of Figure 2e. Importantly, the absolute value of the shift and the width of the magnetic hysteresis loops coincide with those obtained for cooling in negative **H** within the experimental accuracy (Figure 2f,i). These results are a direct manifestation of switching the pinning effect.

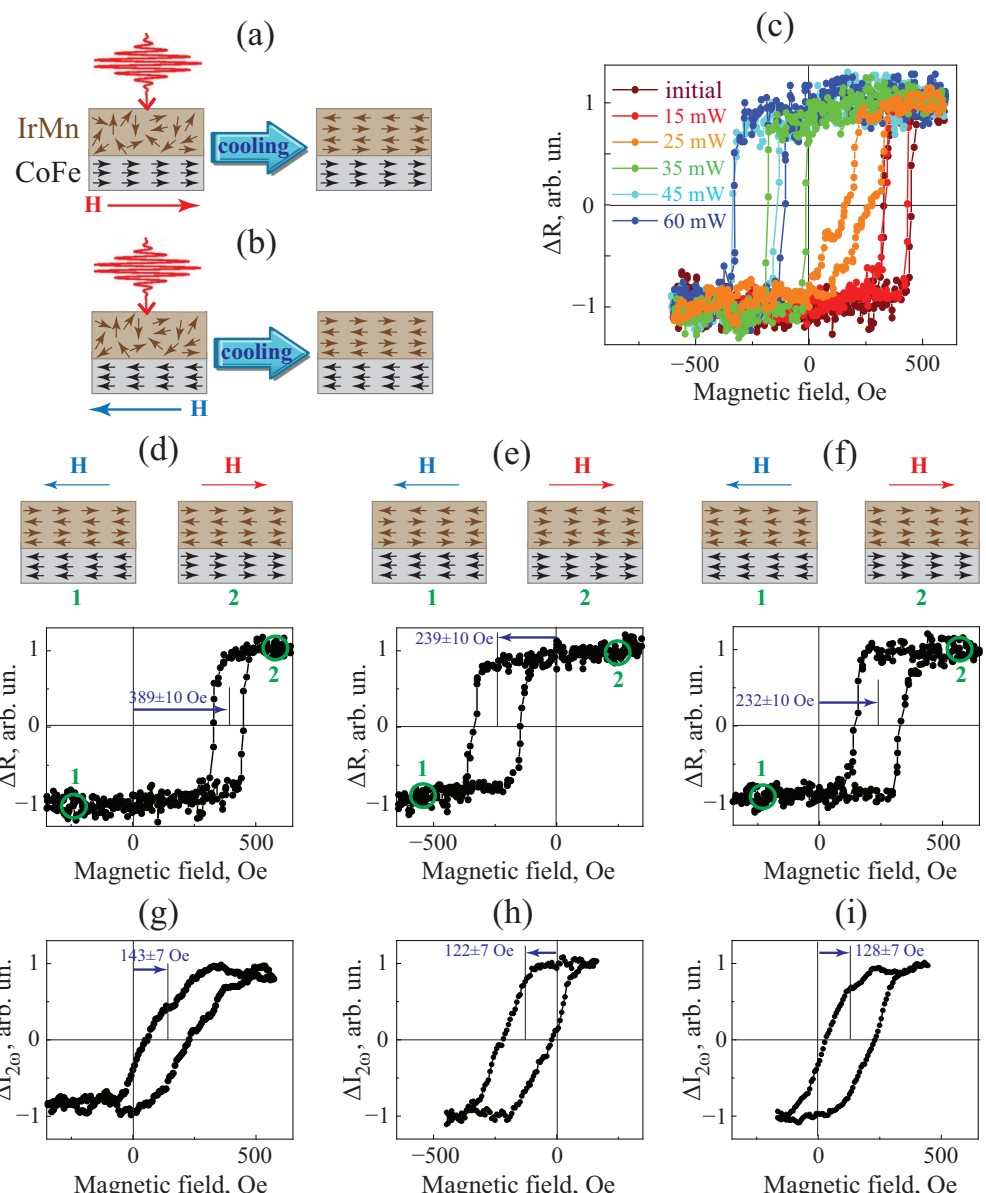

**Figure 2.** (**a**,**b**) Schemes of the pinning switching process; (**c**) MOKE hysteresis loops obtained in an as-prepared IrMn/CoFe film (wine-colored points) and after the laser heating with different light intensities, indicated in the legend. MOKE hysteresis loops obtained in an (**d**) as-prepared IrMn/CoFe film, (**e**) after cooling in the positive magnetic field. and (**f**) after cooling in the negative field. (**g**–**i**) SHG intensity hysteresis loops obtained in the same conditions as in panels (**d**–**f**). Magnetic field was applied parallel to the initial pinning direction and perpendicular to the plane of light incidence, $d_{CoFe} = 4$ nm. All the data were normalized to maximum values.

We pay special attention to the parameters of the hysteresis loops. We found that, for as-prepared films, (i) initial shifts of the centers of the MOKE and SHG hysteresis loops, i.e., $H_{EB}^{MOKE}$ and $H_{EB}^{SHG}$, differ from each other and are $389 \pm 10$ Oe and $143 \pm 7$ Oe,

respectively, and (ii) the widths of the loops are also different and are $120 \pm 10$ Oe and $180 \pm 10$ Oe in the linear and nonlinear response, correspondingly. This difference of hysteresis loops takes place as well in the $100 \times 100$ μm$^2$ film region subjected to heating by the laser beam (Figure 2e,h). Indeed, the $H_{EB}^{MOKE}$ and $H_{EB}^{SHG}$ are $-239 \pm 10$ Oe and $-122 \pm 7$ Oe, respectively, whereas the corresponding loop widths are 180 Oe and 200 Oe.

In order to check whether the switching effect depends on the magnetic anisotropy of the film induced during the films' deposition and annealing, we performed similar experiments as the sample was oriented in such a way that the initial pinning direction was orthogonal to the external magnetic field (Figure 3). In this case, MOKE and SHG hysteresis loops have near-zero width and almost symmetric shape relative to $H = 0$ (Figure 3a,d). Schematic distributions of the spin alignment in two hysteresis points near the saturation are shown in Figure 3a. We revealed that a clear pinning effect appears as a negative shift of both MOKE and SHG hysteresis loops in the film region irradiated by intense laser beam and cooled in the presence of positive magnetic field (Figure 3b,e). In turn, irradiation of the structure in the negative magnetic field is found to induce the pinning effect in the opposite direction (Figure 3c,f). In other words, a laser-induced heating procedure with subsequent cooling in the DC magnetic field allows us to introduce and control the pinning effect independently on the initial orientation of the pinning direction.

Importantly, all the results described above are perfectly reproducible; moreover, laser-induced switching in a single region of the AFM/FM film can be repeated many times back and forth (see Figures S1 and S2 in Supplementary). We note that after such a subjection, the parameters of the magnetic linear and nonlinear hysteresis loops, $H_{EB}^{MOKE}$ and $H_{EB}^{SHG}$, remain stable and the considered structures do not exhibit the training effect, which consists of a decrease in the exchange bias during the magnetic field cycling at hysteresis loop measurements for some of the AFM/FM heterostructures [31] (see Figure S1d in Supplementary).

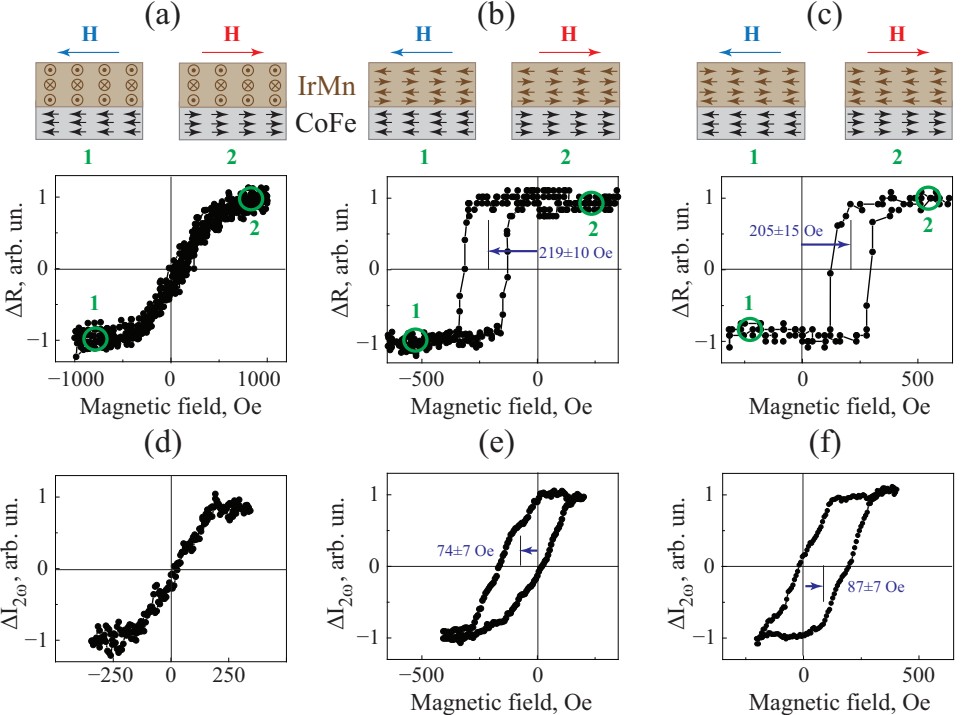

**Figure 3.** MOKE hysteresis loops obtained for (**a**) as-prepared film, (**b**) area of the sample after cooling in the positive field, and (**c**) area of the sample after cooling in the negative field, respectively. (**d**–**f**) SHG intensity hysteresis loops obtained in the same conditions. Magnetic field was oriented perpendicular to the plane of incidence, while the pinning direction lies in the plane of incidence parallel to the film's surface, $d_{CoFe} = 4$ nm. All the data were normalized to maximum (MOKE and SHG) values.

## 4. Discussion

When analysing the MOKE data for the set of samples with different CoFe thickness (Figure 1b), one can see that $H_{EB}$ is nearly inversely proportional to $d_{CoFe}$, as shown in Figure 1d. Such a behavior is expected and consistent with the interfacial nature of the exchange interaction responsible for the pinning effect [32].

The data shown in Figures 2 and 3 clearly illustrate the possibility of switching the pinning effect in AFM/FM films through laser-induced heating of the samples; the role of heating is confirmed by the dependence of the switching parameters on the average intensity of the laser beam and not on the peak intensity. We also show that the suggested procedure of the laser treatment of the AFM/FM films allows one to reorient the pinning direction even to 90 degrees. At the same time, there is a difference between the original values of $|H_{EB}|$ and coercivity (Figure 2d) for as-prepared films and after their laser-induced switching (Figure 2e). This may be attributed to different conditions (temperature, magnetic field) during laser-induced switching compared to those used during the film preparation. Furthermore, laser-induced heating is local and therefore temperature gradients can arise in the illuminated (switching) areas of the film. This may lead to slightly different conditions for switching of the IrMn layer at different points on the sample, which in turn results in broadening the hysteresis loop. Nevertheless, the magnetic hysteresis loops' parameters remain consistent during multiple rounds of laser-induced switching.

Based on the comparison of the linear and nonlinear optical data, we conclude that the shift of the MOKE hysteresis loop relative to zero magnetic field exceeds that of the SHG loop and the strongest difference is observed in the structure with the thinnest CoFe layer of 2 nm (Figure 1d). We note that the linear MOKE characterizes the magnetic properties of the "bulk" of the structure at the penetration depth of light, while the electric dipole SHG sources are localized exclusively at the surface and interfaces of a nanofilm made of a centrosymmetric material. We suppose that just this diversity is responsible for the observed difference of the hysteresis loop parameters of linear and nonlinear magneto-optical response of IrMn/CoFe films. As in the experiment, we detect the averaged SHG response from two CoFe interfaces; the difference in the magnetic properties of these may be important for the observed shift of the SHG hysteresis loop relative to $H = 0$.

The extra contribution to the nonlinear response can be associated with the gradients of the magnetization, $\nabla_z M$, along the normal to the AFM/FM interface, $z$, which results in the difference of the SHG and MOKE hystereses. Indeed, it was found earlier that a nonuniform magnetization can give contributions to the magnetization-induced SHG response [28,29], whereas linear MOKE resulting from the average value of the magnetization of the structure is much less sensitive to it. Obviously, the largest values of $\nabla_z M$ occur in the thinnest sample which sustains our observation of the pronounced difference between $H_{EB}^{SHG}$ and $H_{EB}^{MOKE}$.

It should be noted that the width of the SHG hysteresis loop usually exceeds that of the MOKE loop, while the slope is smaller (see Figure 2d,g). We associate this with roughness of the film interfaces and high sensitivity of the SHG response to the defects of the surface. Analogous effects were observed in polycrystalline FeMn/NiFe bilayers, where areas with different exchange bias fields were introduced by the He ion irradiation [33].

## 5. Conclusions

In conclusion, we studied experimentally the appearance of the switchable pinning effect in IrMn/CoFe films in their linear and nonlinear magneto-optical response. Switching is established as the shift of the magnetic hysteresis loops of the MOKE and SHG response relative to the zero value of the magnetic field applied parallel to the pinning direction. It was found that the hysteresis loop shift for AFM/FM films decreases as the thickness of the ferromagnetic layer increases, as expected for the interfacial exchange interaction in these structures. We revealed a pronounced difference in the shape and parameters of the magnetic hysteresis loops in the linear and nonlinear magneto-optical response of the IrMn/CoFe films; namely, the shift of the SHG hysteresis loop is notably smaller than

that of the MOKE. This distinction increases with decreasing CoFe thickness which can be associated with gradients of magnetization at the IrMn/CoFe interface and the roughness of the surfaces.

**Supplementary Materials:** The following supporting information can be downloaded at: https://www.mdpi.com/article/10.3390/photonics10121303/s1, Figure S1: MOKE hysteresis loops for as-prepared and multiple switched pinned films; Figure S2: SHG hysteresis loops for as-prepared and multiple switched pinned films.

**Author Contributions:** Conceptualization, T.V.M. and I.A.K.; methodology, V.B.N.; formal analysis, E.A.K.; sample preparation, N.S.G. and I.Y.P.; investigation, I.A.K. and V.B.N.; writing—original draft preparation, I.A.K.; project administration, T.V.M. All authors have read and agreed to the published version of the manuscript.

**Funding:** This work is supported by the Russian Science Foundation, grant No. 19-72-20103.

**Data Availability Statement:** The data presented in this study are available on request from the corresponding author.

**Conflicts of Interest:** The authors declare no conflict of interest.

## Abbreviations

The following abbreviations are used in this manuscript:

| | |
|---|---|
| SHG | second harmonic generation |
| MOKE | magneto-optical Kerr effect |
| PM | power-meter |
| PMT | photomultiplier tube |
| AFM | antiferromagnet |
| FM | ferromagnet |

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
