# Peer review of "Second Harmonic Generation Versus Linear Magneto-Optical Response Studies of Laser-Induced Switching of Pinning Effects in Antiferromagnetic/Ferromagnetic Films"

_photonics, doi:10.3390/photonics10121303_

Round 1

Reviewer 1 Report (Previous Reviewer 1)

Comments and Suggestions for Authors

No

Author Response

There were no Comments and Suggestions for Authors from Reviewer 1.

We did our best to correct all the typos in the text of the manuscript.

Reviewer 2 Report (Previous Reviewer 2)

Comments and Suggestions for Authors

All my comments have been addressed adequately. In my opinion, the paper can be accepted, keeping in mind it is just a collection of results (linear MOKE vs SHG), without any deep explanation/interpretation of the involved phenomena.

Comments on the Quality of English Language

Still typos in the text.

Author Response

We are thankful to the Reviewer for the positive opinion on our manuscript.

We did our best to correct all the typos in the text of the manuscript, the changes in the text are marked in the .diff file.

Unfortunately, the changes in the Abstract are not seen - we excluded the last sentence from the previous version, as it nearly repeated the previous one.

This manuscript is a resubmission of an earlier submission. The following is a list of the peer review reports and author responses from that submission.

Round 1

Reviewer 1 Report

Comments and Suggestions for Authors

The authors describe experimental results on the change of the unidirectional anisotropy in an EB layer system after fs-laser irradiation.

In principle, the data are poorly presented and far from adequately discussed. In addition to some errors, such as unaxial in place of undirectional, no time-dependent dynamics are shown, and likewise no optical Kerr-microscope images showing the interaction region with the laser - especially with repeated magnetic field changes. The thickness dependence of the EB effect of CoFe must be strictly linear, which is not seen in Fig. 1d.

In principle, the results are not new; results exist from 20 years ago that showed this effect. Also the literature is not up to date - e.g. a paper from the arxiv is missing: Guo et al.- laser induced switching of an exchange bias antiferromagnet.

There would be more to complain about, so I think a publication without significant change and extension of the presentation, especially without representative time-dependent results, is out of the question.

Author Response

Reply to the Reports of the Reviewers –

Photonics-2657198

I.A. Kolmychek et al.,

Laser-Induced Switching of Pinning Effects in Antiferromagnetic/Ferromagnetic Films: Second Harmonic Generation Versus Linear Magneto-Optical Response Studies

Dear Reviewers, dear Editor,

We are thankful for a thorough reading of our manuscript and useful comments.

We have considered critically the manuscript in order to meet all recommendations of the Reviewers and hope that it made it more clear and appealing.

Point by point reply to the comments of the Reviewers (in Italic) are given below.

We also prepared a version of manuscript (file “diff.pdf”) with highlighted changes.

Reviewer 1

Reviewer:

In principle, the data are poorly presented and far from adequately discussed. In addition to some errors, such as unaxial in place of undirectional, no time-dependent dynamics are shown, and likewise no optical Kerr-microscope images showing the interaction region with the laser - especially with repeated magnetic field changes.

Authors’ reply:

We are sorry on such a critical opinion of the Reviewer on our manuscript and would like to clearify some issues.

  • There are some typos in the text, sorry for this; we made the necessary corrections.
  • We did not intend to study the time dependent dynamics, it was studied in much detail e.g. in the paper Saito, Y.; Kholid, F.N.; Karashtin, E.; Pashenkin, I.; Mikhaylovskiy, R.V. Terahertz Emission Spectroscopy of Exchange-Biased Spintronic Heterostructures: Single- and Double-Pump Techniques. Rev. Appl. 2023, 19, 064040. https://doi.org/10.1103/PhysRevApplied.19.064040 cited in the manuscript. Similarly, we did not intend to study the dynamics of magnetization using the Kerr microscopy technique – this would be another paper and another aim. We also do not pretend to suggest the optimal and the most efficient structure for possible applications for ultrafast spintronic devices.
  • For us, the goal of this manuscript was the comparative study of the appearance of the linear and nonlinear-optical response of the systems that reveal the laser-induced switching of the pinning effect. And the idea here was to compare the switching behavior of the bulk and the surface of the IrMn/FM nanolayers, as the second harmonic probe is specific mostly to the properties of the interfaces in contrast to the linear optical response. Perhaps this idea was not clearly presented in the text; we did our best to clear up in the updated version of the manuscript.
  • Speaking about the suggested Kerr microscopy studies – we suppose that such an experiment was not necessary for our studies, as we checked many times that the illuminated region of the film revealed the same switching properties at the scales of a few dozens of microns, i.e. at the laser spot region. This seems to be enough to confirm that the film was not damaged. It seems also that the experiments if the Kerr microscopy are not the necessary requisite for publishing a paper.

Reviewer:

The thickness dependence of the EB effect of CoFe must be strictly linear, which is not seen in Fig. 1d.

Authors’ reply:

Perhaps there is a misunderstanding here, as the dependence of the EB effect should decrease as the thickness of the ferromagnetic layer grows, namely the EB should be inversely proportional to the CoFe thickness H_EB~1/d_CoFe, as it appears in Fig. 1d (see, i.e. [J. Nogués et al. J. of Magnetism and Magnetic Materials 192, 203 (1999)] or [R.L. Stamps, J. Phys. D: Appl. Phys. 33, R247 (2000)]). There is no mistake in this graph.

Reviewer:

- In principle, the results are not new; results exist from 20 years ago that showed this effect.

Authors’ reply:

  • This statement depends of what effect is considered. If the Reviewer mentions the EB switching, we agree. While the idea of the direct comparison of the linear (i.e. bulky) and second-order nonlinear (interface-sensitive) effect has not been considered up to now to the best of our knowledge. Moreover, the studies of the EB switching by the SHG probe has not been studied either. In the current version of the manuscript, we did our best to figure our this main aim of the manuscript.

Reviewer:

- Also the literature is not up to date - e.g. a paper from the arxiv is missing: Guo et al.- laser induced switching of an exchange bias antiferromagnet.

Authors’ reply:

  • We are thankful to the Reviewer for this comment and apologize for missing this ArXiV paper. We included this suggested paper and additional references [7, 8,19, 20, 21, 22, 23] to the list of references.

Action taken:

The reference to ArXiV paper was added (reference [12] in the revised version of manuscript). As well as the references [7, 8,19, 20, 21, 22, 23] were added.

Reviewer:

  • There would be more to complain about, so I think a publication without significant change and extension of the presentation, especially without representative time-dependent results, is out of the question.

Authors’ reply:

  • We are thankful to the reviewer for this criticism and paving our attention to the most principle points that should have been improved.

Action taken:

            We made the corrections in the title,  abstract, conclusions and in the main text of the manuscript to emphasize the main goal of the manuscript: to compare the switching behavior of the bulk and the surface of the IrMn/FM nanolayers, probed by MOKE and SHG, respectively.

Reviewer 2 Report

Comments and Suggestions for Authors

The manuscript by I. A. Kolmychek et al. reports a detailed characterization of the magnetic properties of IrMn/CoFe films when subjected to a laser heating treatment, in order to modify and control the exchange bias effect in such a system. The study is systematic (e.g. they consider various CoFe thicknesses and various laser fluences), and both linear transverse MOKE and SHG have been employed. Nonetheless, I failed in finding what is the main message of the manuscript, and I think this is a major issue to be addresses before publication. From the abstract I can read: "In this work we demonstrate laser-induced switching of the pinning direction via thermal-induced break down of the exchange bias effect in IrMn/CoFe films". Is this the main claim of the work? If that is the case, I'm afraid there are already several works investigating the same aspect. Few examples (not cited):

1) "Manipulating exchange bias using all-optical helicity-dependent switching", Phys. Rev. B 96, 144403 (2017). In this work, the authors also consider the role of the helicity of the heating laser.

2) "Exchange Bias Realignment Using a Laser-based Direct-write Technique", Physics Procedia 56, 1136 (2014). In this work, the authors did a similar study as the submitted manuscript, while scanning the laser in a direct-writing fashion.

3) "Exchange bias and diffusion processes in laser annealed CoFeB/IrMn thin films", J. Magn. Magn. Mater 489, 165390 (2019). Another study on laser annealing.

I do think the submitted work is worth to be published (the comparison between linear MOKE and SHG is interesting), but in my view a clear definition of the goal of the article is crucial, especially considering the existing literature.

Moreover, I list here a few minor comments:

1) Why did the authors use broadband light for linear MOKE and not a low-power CW laser? Are there advantages of using a broad spectrum (without doing spectroscopy)?

2) Why is the 4 nm loop (green) in Fig. 1(c) opposite with respect to the others? As the authors commented, this also reflects in the 1(e) panel, where the 4 nm point for the SHG measurement is positive. Could this be an artifact? Is it related to the phase of the magnetic response?

3) In section 3.2, the authors have chosen the 4 nm sample as the representative case. Why is it so? The 4 nm case is the only one falling out of the trend in Fig. 1 (see point 3 above), so it doesn't seem the best representative case here.

4) The caption of Fig. 2 is missing the CoFe thickness, which is a relevant info.

5) For the laser heating, what is the exposure time? Is it the same for all the fluences?

6) The paragraph starting with "Importantly..." (line 248) contains two important messages: The fact the laser-induced switching is repeatable and the absence of training effect. It would be very useful to see a sequence of measurements as supplementary info, to support the two claims.

Comments on the Quality of English Language

I found several typos, please have a look at them (e.g. "interation" to be substituted with "interaction", ...).

Author Response

Reviewer 2:

Reviewer:

The manuscript by I. A. Kolmychek et al. reports a detailed characterization of the magnetic properties of IrMn/CoFe films when subjected to a laser heating treatment, in order to modify and control the exchange bias effect in such a system. The study is systematic (e.g. they consider various CoFe thicknesses and various laser fluences), and both linear transverse MOKE and SHG have been employed. Nonetheless, I failed in finding what is the main message of the manuscript, and I think this is a major issue to be addresses before publication. From the abstract I can read: "In this work we demonstrate laser-induced switching of the pinning direction via thermal-induced break down of the exchange bias effect in IrMn/CoFe films". Is this the main claim of the work? If that is the case, I'm afraid there are already several works investigating the same aspect.

Authors’ reply:

  • We are thankful to the Reviewer for this very important criticism. Definitely, the idea of the manuscript was mostly the comparison of the appearance of the linear and nonlinear-optical (SHG) response of the systems that reveal the laser-induced switching of the pinning effect, i.e. to compare the switching behavior of the bulk and the surface of the IrMn/FM nanolayers, as the second harmonic probe is specific mostly to the properties of the interfaces in contrast to the linear optical response. We made the corresponding corrections in the abstract and in the main text of the manuscript.

Action taken:

We made corrections in the title, abstract, conclusions and main text of the manuscript to emphasize the main idea of the paper.

Reviewer:

Few examples (not cited):

1) "Manipulating exchange bias using all-optical helicity-dependent switching", Phys. Rev. B 96, 144403 (2017). In this work, the authors also consider the role of the helicity of the heating laser.

2) "Exchange Bias Realignment Using a Laser-based Direct-write Technique", Physics Procedia 56, 1136 (2014). In this work, the authors did a similar study as the submitted manuscript, while scanning the laser in a direct-writing fashion.

3) "Exchange bias and diffusion processes in laser annealed CoFeB/IrMn thin films", J. Magn. Magn. Mater 489, 165390 (2019). Another study on laser annealing.

Authors’ reply:

- the authors are thankful to the Reviewer for these suggestions, we included these papers to the list of references.

Action taken:

We added the mentioned references (references [21], [22] and [23] in the revised version of the paper) and several other important works concerning the EB switching (references [7], [8], [12], [19], [20] in the revised version).

Reviewer:

I do think the submitted work is worth to be published (the comparison between linear MOKE and SHG is interesting), but in my view a clear definition of the goal of the article is crucial, especially considering the existing literature.

Authors’ reply:

- the authors are thankful to the Reviewer for this positive opinion on the manuscript and results presented in it. We did our best to make the main message of the manuscript more clear.

MINOR COMMENTS

Reviewer:

1) Why did the authors use broadband light for linear MOKE and not a low-power CW laser? Are there advantages of using a broad spectrum (without doing spectroscopy)?

Authors’ reply:

- We performed the MOKE studies with different light sources, while it turned out that the lowest noise was for the thermostabilized lamp, and that was the advantage.

We also performed the MOKE measurements using the low-power CW laser and found that the results in that case are similar to those obtained using the lamp. This is underlined as well in the text of the manuscript, line 150. We also add the results of the corresponding measurements to the initial text of the manuscript - the dependence of the exchange bias field on the CoFe thickness measured with the use of the radiation of the Ti-Sa laser operating in the CW mode is shown in Fig. 1d, blue points.

Reviewer:

2) Why is the 4 nm loop (green) in Fig. 1(c) opposite with respect to the others? As the authors commented, this also reflects in the 1(e) panel, where the 4 nm point for the SHG measurement is positive. Could this be an artifact? Is it related to the phase of the magnetic response?

Authors’ reply:

- Actually, the shape of the magnetization-induced SHG loops for the samples with different CoFe thickness were measured several times, including the structure with 4 nm CoFe thickness, all the results are reproducible, so this cannot be an artifact.

 Apparently, the fact that the 4 nm loop is opposite with respect to the others should be attributed to the result of the interference of the magnetization-induced and crystallographic (nonmagnetic) contributions of the CoFe\IrMn and CoFe\glass interfaces, which we suppose are the main nonlinear sources responsible for the magnetization-induced effect in this system.

We agree with the Reviewer that the phase of the magnetic response plays a crucial role here, as it is noted in the initial version of the manuscript.   

Reviewer:

3) In section 3.2, the authors have chosen the 4 nm sample as the representative case. Why is it so? The 4 nm case is the only one falling out of the trend in Fig. 1 (see point 3 above), so it doesn't seem the best representative case here.

Authors’ reply:

- We performed analogous experiments for all the samples and obtained similar results. The choice of the structure with the CoFe thickness of 4 nm was argued by two reasons: (i) from the one hand, this sample was expected to provide high value of the EB field (for the sample with CoFe thickness 8 nm the H_EB is twice lower), which is of the interface nature,  and (ii) from the other hand, the structure with d_CoFe=2 nm is too thin, thus it is hard to distinguish the “bulk” contribution to magnetization probed by MOKE and “interface” contribution probed by the SHG.  

Reviewer:

4) The caption of Fig. 2 is missing the CoFe thickness, which is a relevant info.

Authors’ reply:

- Thank you for the comment, we added to the captions of Fig.2 and Fig.3 that d_CoFe=4 nm.

Action taken:

We added to the captions of Fig.2 and Fig.3 that d_CoFe=4 nm.

Reviewer:

5) For the laser heating, what is the exposure time? Is it the same for all the fluences?

Authors’ reply:

The exposure time was the same for all the experiments and was about 30 msec. In our measurements we had to study relatively large areas of the sample of about 30 μm in scale, while the exposure region was 100 μm x 100 μm. To provide homogeneous exposition of the switching laser radiation a special algorithm was applied, which scanned this region by the waist of focused heating beam. The the waist translation speed was about 1 mm/sec that corresponds to about 30 msec of exposure. For mean light intensity of 8.5 kW/cm2 the resulted fluence is 250 J/cm2.

Reviewer:

6) The paragraph starting with "Importantly..." (line 248) contains two important messages: The fact the laser-induced switching is repeatable and the absence of training effect. It would be very useful to see a sequence of measurements as supplementary info, to support the two claims.

Authors’ reply:

  • We are thankful to the Reviewer for this query. Really, the repeatability of switching is just postulated in the text of the manuscript. In accordance with your request, we made the supplementary materials to this manuscript. We introduced there (1) the illustration on how the MOKE hysteresis loops appear in consequent measurements of the EB switching under the application of the positive and negative magnetic field (Supplementary file, Fig. S1), as well as the magnetic hysteresis loops of the SHG intensity (Supplementary file, Fig. S2); and (2) a consequent set of the linear MOKE hysteresis loops for the same EB state, which is an illustration of the training effect (Supplementary file, Fig. S1 (d)). For the SHG results the training effect was not studied, as each SHG measurement was quite a long and typically took dozens of minutes. Each single point for fixed magnetic field was accumulated for 1 sec., and in order to get reliable as we had to average the data over 10-20 measurements. Each of them was relatively noisy and did not allow to reveal the training effect – especially if taking into account that the effect is negligibly small, which stems from the linear-optical MOKE studies.

Action taken:

We made a Supplementary file to show the absence of the training effect and the reproducibility of the laser switching of the EB.
